# Intra-Amniotic Inflammation or Infection: Suspected and Confirmed Diagnosis of “Triple I” at Term

**DOI:** 10.3390/children10071110

**Published:** 2023-06-26

**Authors:** Sara Consonni, Elettra Salmoiraghi, Isadora Vaglio Tessitore, Armando Pintucci, Valentina Vitale, Patrizia Calzi, Francesca Moltrasio, Anna Locatelli

**Affiliations:** 1Department of Obstetrics and Gynecology, Carate Hospital, ASST Brianza, 20871 Vimercate, Italy; sara.consonni@asst-brianza.it (S.C.); isadora.vagliotessitore@asst-brianza.it (I.V.T.); 2School of Medicine and Surgery, University of Milano-Bicocca, 20126 Milan, Italy; e.salmoiraghi1@campus.unimib.it (E.S.); v.vitale10@campus.unimib.it (V.V.); 3Department of Obstetrics and Gynecology, Desio Hospital, ASST Brianza, 20871 Vimercate, Italy; armando.pintucci@asst-brianza.it; 4Department of Pediatrics, Carate Hospital, ASST Brianza, 20871 Vimercate, Italy; patrizia.calzi@asst-brianza.it; 5Department of Pathology, Desio Hospital, ASST Brianza, 20871 Vimercate, Italy; francesca.moltrasio@asst-brianza.it; 6Obstetrics, Fondazione IRCCS San Gerardo dei Tintori, 20900 Monza, Italy

**Keywords:** triple I, chorioamnionitis, intra-amniotic infection, intra-amniotic inflammation, intrauterine infection, placental histology, early neonatal sepsis

## Abstract

Chorioamnionitis (CA) at term of pregnancy can have an infectious and/or inflammatory origin and is associated with adverse outcomes. Triple I (intrauterine inflammation, infection, or both, TI) has been proposed to reduce the overdiagnosis of infection and neonatal overtreatment. The aim of this study is to identify clinical and histological variables that could predict adverse outcomes when TI is suspected and/or confirmed. This retrospective cohort study included 404 pregnancies (gestational age ≥ 37 weeks) that were divided into 5 all-inclusive and mutually exclusive groups. TI was defined according to the NICHD definition of 2015, and it could be confirmed (TI+) or not confirmed (TI−) via histological examination. Signs of infection/inflammation that did not conform to the definition of TI were classified as “clinical suspicion” and could be supported (CS+) or not supported (CS−) by histology. Cases of histological chorioamnionitis (HCA) without clinical manifestation represented a fifth group. Whole placental involvement (WPLI) was defined as a histological inflammation involving the maternal and fetal sides. There were 113 TI+, 30 TI−, 186 CS+, 35 CS−, and 40 isolated HCA cases. WPLI was diagnosed in 133 cases (39.2%). Composite neonatal outcome (CNO) occurred in 114 cases (28.2%) while composite maternal outcome (CMO) occurred in 192 cases (47.5%). Compared with CS+, TI+ was more predictive of CNO (*p* = 0.001), CMO (*p* < 0.001), and WPLI (*p* = 0.005). WPLI was related both to CNO (*p* < 0.001) and to CMO (*p* = 0.046). TI+ and WPLI showed similar sensitivity but different specificity in predicting CNO. At logistic regression, CNO was independently predicted by TI+ (OR 2.21; *p* = 0.001) and by WPLI (OR 2.23; *p* = 0.001). Compared with CS, TI is a better predictor of CNO and can be useful for the identification of newborns at risk.

## 1. Introduction

In 2015, an expert panel convened by the National Institute of Child Health and Human Development recommended the use of the term “triple I” or TI (intrauterine inflammation, infection, or both) in place of the term chorioamnionitis in order to refine the definition and management of a heterogeneous array of conditions characterized by infection and inflammation in pregnancy [1]. The TI definition was suggested because single clinical variables are not able to predict clinical outcomes and have a low diagnostic performance if considered alone, whereas in combination they seem to more reliably predict both maternal and fetal outcomes. Maternal fever, which is the main diagnostic criterion, has a very low diagnostic accuracy if considered alone, since it can be caused by extra-uterine infection/inflammation, by the use of epidural anesthesia, or by prostaglandins used to induce labor. Fetal tachycardia may also be present in the absence of chorioamnionitis; it may for example be related to medications (e.g., ephedrine and beta agonists) or to transient fetal hypoxia [2]. Maternal fever and fetal tachycardia together have been described as risk factors for adverse maternal and neonatal outcomes [3,4,5]. Maternal leukocytosis has a low specificity (5–30%) as a sign of intrauterine infection (IUI) since it can be caused by corticosteroid administration, labor pain, and labor duration [6]. Maternal leukocytosis on admission can even be associated with severe adverse infant neurodevelopmental outcomes [7]. Meconium-stained and/or foul-smelling amniotic fluid seem to be more specific signs, and, in particular, they can predict funisitis in cases of maternal fever [8]. Purulent amniotic fluid is more frequent in patients with severe or prolonged infections [9]. Both of these conditions should alert the physician to the potential for infection and increased perinatal morbidity [10].

“Histological chorioamnionitis” (HCA) refers to histopathological inflammation with or without clinical or microbiological findings associated with acute infection, introducing the possibility of the sterile inflammation of tissues [11]. The prevalence of HCA is inversely related to gestational age, occurring in 20% of deliveries at term after spontaneous labor and in up to 50% of preterm births [12]. Histological data retrospectively support the diagnosis of triple I according to the definition of this condition. The association between placental histopathologic findings of acute inflammation and adverse clinical outcomes has been described in the literature [13]. The deepness of placental invasion by polymorphonucleated cells defines the grading and staging of HCA, which are related to the severity of the process and consequently to adverse maternal and neonatal outcomes. An advanced HCA stage leads to a worse prognosis, such as the occurrence of neonatal sepsis. Maternal fever in term deliveries worsens maternal and neonatal outcomes in the presence of HCA [14,15]. Even worse perinatal outcomes (e.g., respiratory distress syndrome, neonatal sepsis, pneumonia, necrotizing enterocolitis) can occur if the inflammatory response involves the umbilical vein, the arteries, and the chorionic plate surface vessels (the funisitis scenario) [16,17,18]. On the maternal side, HCA, and in particular funisitis, are associated with labor dystocia, postpartum hemorrhage, postpartum fever, endometritis/myometritis, and even sepsis [19,20,21].

Some studies have been investigating the role of TI criteria in the diagnosis of IUI, with controversial results; therefore, more studies about the clinical implications of the definition of TI need to be performed [22,23]. In particular, it is unclear whether the clinical signs or the histology pattern better relate to adverse perinatal and maternal outcomes, and how the combination between the two can impact maternal and neonatal prognoses.

The primary aim of our study is to assess the relationship between clinical antenatal signs that support or do not support the complete TI spectrum and perinatal and maternal outcomes. The second aim is to evaluate the relationship between the clinical criteria of TI and placental histology, and to investigate the ability of placental histologic findings to predict perinatal outcomes.

## 2. Materials and Methods

### 2.1. Population

We conducted a retrospective cohort study that included term pregnancies (≥37 weeks) with a clinical suspicion or histological diagnosis of IUI or inflammation. The cases used in the study originated from a histological analysis of all the term placentas sent to the Pathology Unit because of a clinical suspicion of IUI, or for other reasons (Figure 1), in the period January 2014–July 2017.

The inclusion criteria were as follows: gestational age ≥ 37 weeks and a request for placental histological analysis due to clinical signs of chorioamnionitis or other indications. The exclusion criteria were as follows: deliveries which did not have a histopathologic placental exam and delivery before 37 weeks.

This study was approved by the Ethic Committee of Monza e Brianza (approval code: 1909, 30 October 2014).

The included cases were divided into diagnostic categories based on clinical and histological findings (Figure 1).

The following variables were collected for suspected and confirmed TI: temperature, white blood cell count, fetal heart rate, qualitative amniotic fluid characteristics, and placental histology (which could show acute inflammation of the maternal compartment, the fetal compartment, or both). We also collected the following data: maternal age, body mass index (BMI), parity, gestational age, group B Streptococcus (GBS) results, occurrence of premature rupture of membranes (PROM), time from PROM to delivery, induction of labor, use of oxytocin, epidural anesthesia, duration of labor, number of vaginal examinations, mode of delivery (vaginal or caesarean), blood loss, and antibiotic therapy during labor and in the postpartum period.

In accordance with the NICHD definition, the TI diagnostic criteria were a maternal temperature of 38.0 °C or greater plus any of the following:-Fetal tachycardia (greater than 160 beats per min for 10 min or longer, excluding accelerations, decelerations, and periods of marked variability);-A maternal white blood cell count greater than 15,000 per mm^3^ in the absence of corticosteroids;-Definite purulent fluid from the cervical os.

Based on clinical and histological findings, the study population was divided into the following five all-inclusive and mutually exclusive groups:Suspected TI (TI−): cases with clinical signs fulfilling all TI diagnostic criteria but without confirmation via histological analysis.Confirmed TI (TI+): cases with clinical signs fulfilling all TI criteria plus positive placental histology of infection or acute inflammation.Clinical suspicion with positive histology (CS+): suspected cases not fulfilling all TI diagnostic criteria with positive placental histology (i.e., isolated maternal fever or maternal leukocytosis with meconium-stained amniotic fluid).Clinical suspicion with negative histology (CS−): suspected cases not fulfilling all TI criteria with negative placental histology.Histological chorioamnionitis without clinical signs (HCA): cases of acute inflammation at placental histological examination but without fever or symptoms or signs of chorioamnionitis.

The composite maternal outcome included blood loss > 1000 mL, postpartum fever, endometritis, postpartum antibiotic therapy, or hospital re-admission.

The composite neonatal outcome included an Apgar score at 5 min < 7, C-reactive protein (CRP) > 1 mg/dL, early onset sepsis (EOS), antibiotic therapy at discharge, or NICU admission.

### 2.2. Placental Analysis

Placental samples were collected and analyzed at the Pathology Unit of ASST-Brianza according to the classification system proposed by Redline, in which placental lesions are divided into maternal and fetal inflammatory responses (stages 1, 2, or 3) [14]. On the maternal side, stage 1 is defined by the presence of neutrophils in the subchorial intervillous space or beneath the chorion laeve layer (acute subchorionitis or chorionitis); in stage 2, the involvement of both the chorion and amnion is observed; and in stage 3, necrotizing chorioamnionitis is observed. In fetal inflammatory response, stage 1 involves umbilical phlebitis; in stage 2, the involvement of the umbilical vein and one or more umbilical arteries is observed; and in stage 3, necrotizing funisitis is observed. We defined WPLI (whole placental involvement) as a concurrent inflammatory response of the maternal and fetal side of the placenta (acute chorioamnionitis plus fetal inflammatory response).

### 2.3. Statistical Analysis

Categorical variables were reported as numbers and percentages, and continuous variables were reported as means and standard deviations. The categorical variables were analyzed using Chi-square tests, and the continuous variables were analyzed using ANOVA tests. A *p*-value < 0.005 or an odds ratio (OR) with a 95% CI not inclusive of the unity was considered statistically significant.

The sensitivity, specificity, positive predictive values (PPVs), negative predictive values (NPVs) and likelihood ratios (LRs) of TI and WPLI as predictive factors of composite neonatal outcome were calculated.

A multivariate logistic regression was performed to assess the independent effect of histological damage and clinical signs on composite adverse neonatal outcome.

Statistical analyses were performed using SPSS version 22.0.

## 3. Results

Between January 2014 and July 2017 in the Maternity Unit of Carate Brianza Hospital, Vittorio Emanuele III, out of 6962 term deliveries, 1122 placentas were sent for histopathological analysis. The study population was composed of 404 term deliveries (364 cases presenting symptoms or signs suggestive of inflammation/infection, plus 40 cases of isolated HCA without clinical signs).

Within the cases with clinical signs suggestive of infection or inflammation, 299 placentas (82.1%) presented positive histological findings, while 65 (17.9%) were negative.

The study population was divided into five groups as follows (Figure 2 and Figure 3): TI+ (*n* = 113, 28%), TI− (*n* = 30, 7%), CS+ (*n* = 186, 46%), CS− (*n* = 35, 9%), and HCA (*n* = 40, 10%). 

Among the 339 placentas (30.2%) presenting histological chorioamnionitis, 204 (60.2%) presented only maternal inflammatory response, 2 (0.6%) presented only fetal inflammatory response, and 133 (39.2%) presented WPLI. 

The main characteristics of the study population are summarized in Table 1.

Antibiotic therapy during labor was administered to 60.5% cases: ampicillin was used in 31.3% of cases, amoxicillin + clavulanic in 12.9% of cases, and broad-spectrum antibiotics were used in 16.4% of cases. MSAF was observed in 51% of cases, of which 82% had a suspected IUI confirmed via histology. Excluding cases of HCA without clinical suspicion, the presence of MSAF was very common (56.5%), and it was equally frequent in cases with negative vs. positive histology findings for HCA (67% vs. 54%, *p*-value = 0.13).

The maternal and neonatal outcomes are described in Table 2. Composite maternal adverse outcomes occurred in 192 patients (47.5%). Neonatal composite adverse outcomes occurred in 114 cases (28.2%), and early onset sepsis occurred in only 4 cases (0.9%), all of which were TI+ cases. None of the newborns were re-hospitalized after discharge. Interestingly, in TI+ and TI− cases, the incidence of maternal and neonatal composite outcome was similar, except for EOS, which occurred only in TI+.

Involvement of the maternal side of the placenta occurred with a similar frequency (98.2% vs. 98.9%) in TI+ and CS+, while fetal side inflammation was present in 51.3% of TI+ cases and in 35.5% of CS+ cases (*p*-value = 0.008). Comparing TI+ with CS+, TI+ was more closely related to WPLI (51.3% vs. 34.4%; *p*-value = 0.005) and to neonatal (45.1% vs. 25.3%, *p*-value = 0.001) and maternal composite outcomes (72.6% vs. 36.6%; *p*-value < 0.001). WPLI was significantly related both to maternal (37.0% vs. 29.2%; *p*-value = 0.046) and neonatal composite adverse outcomes (47.4% vs. 27.2%; *p*-value < 0.001).

At logistic regression, both WPLI and TI+ proved to be independent predictors of neonatal composite adverse outcome, with respective OR values of 2.23 and 2.21 (Table 3).

The sensitivity, specificity, positive and negative predictive values and likelihood ratios were calculated for both TI+ and WPLI to assess their ability to predict adverse neonatal outcomes in the context of IUI. As is shown in Table 4, TI had better diagnostic accuracy than histopathologic WPLI.

## 4. Discussion

Early diagnosis of IUI is crucial to reduce adverse outcomes. However, isolated antenatal clinical signs do not show reliable diagnostic accuracy [22]. Hence, the proposed use of the TI criteria to diagnose IUI and to predict neonatal and maternal adverse outcomes [1]. Although the TI criteria were introduced by the NICHD in 2015, they have not been implemented in clinical practice, and several recent studies have questioned their utility. Ona S. et al. [24] concluded that applying the TI criteria to guide the clinical diagnosis of IUI or inflammation may misdiagnose women at risk for adverse infectious outcomes since the sensitivity and specificity of suspected TI for an adverse infectious outcome were only 67.6% and 38.1%, respectively. Their study included several limitations, such as including preterm pregnancies, possible selection bias, and a lack of placental examination data.

In contrast, our study revealed that both TI+ and TI− are better predictors of adverse outcomes at term than CS. In particular, when compared with CS, only TI+ showed a significant statistical association with both clinical adverse maternal and neonatal outcomes as well as TI and more extensive placental involvement (WPLI). Therefore, even in cases of negative placental histology, the clinical criteria of TI can be useful as a predictive test. Furthermore, all four EOS cases—the most significant outcome—occurred in the TI+ group. It is of interest that in our series, the incidence of EOS at term was similar to that reported in the literature (0.1% vs. 0.04%) [25].

A recent review on the subject has recommended the implementation of the TI criteria according to the current clinical guidelines so as to avoid the overexposure of newborns to broad-spectrum antibiotics, which have potential short- and long-term adverse effects. Neonatal management should be guided by the clinical signs (isolated maternal fever, suspected triple I, or confirmed triple I). In cases of isolated fever and suspected TI before 34 weeks or in cases of confirmed triple I, neonates should be treated, while in cases of isolated maternal fever or cases of suspected TI after 34 weeks, the neonate’s clinical condition should be evaluated. In this way, well-appearing term and late preterm neonates who are asymptomatic can be closely observed without antibiotics [26].

Placental histology is an important tool for improving perinatal care, for establishing a proper classification system, and for providing adequate and reliable diagnoses [27]. Information from placental histology can be useful for confirming a diagnosis of IUI and for predicting early and late adverse outcomes. Our analysis suggests that in the context of suspected IUI, even in the absence of severe signs that would lead to a TI diagnosis, WPLI independently predicts adverse perinatal outcomes. Therefore, a placental analysis could be incorporated to differentiate at-risk cases.

Some authors have investigated the utility of frozen placental sections for the fast and early diagnosis of chorioamnionitis as a sensitive screening method which can be performed directly in the delivery room by pathologists or trained clinicians [28,29]. The results of this technique can be obtained in less than 20 min, making it much faster than the traditional histopathologic method which requires 7 days. Hence, fast placental analysis in cases with clinical signs that do not match all of the TI criteria could provide reliable information to drive clinical management. In our study, HCA occurred in 9.9% of cases, a prevalence consistent with previous reports (around 10%) [10], confirming the reliability of our analysis.

We have found that WPLI and TI+ are independent predictors of adverse neonatal outcomes, each one doubling the likelihood of such an outcome. However, TI+ had better diagnostic indices than WPLI.

A recent study, there results of which were at variance with our findings, found that the sensitivity of the TI criteria in the prediction of clinical adverse infectious outcomes was higher than its specificity (67.6% vs. 38.1%) [24]. Furthermore, TI+ appeared to be related to a more frequent fetal placental involvement and maternal adverse outcomes compared with clinical suspicion of IUI and/or cases of inflammation that did not fully conform with the TI diagnostic criteria.

MSAF is a predictor of increased perinatal and neonatal morbidity and mortality, and it can be caused by an adverse intrauterine environment, including chorioamnionitis. As MSAF is not rare (it is present in 8–20% of term deliveries), special attention should be paid to term deliveries with MSAF when associated with strong predictors of complications, such as intrapartum fever, polyhydramnios, gestational diabetes, and fetal heart rate alterations [30]. In our population, when excluding cases of HCA without clinical suspicion, the presence of MSAF was very common (56.5%), and it was equally frequent in cases with negative vs. positive histology (67% vs. 54%, *p*-value = 0.13). Other studies have reported that histologic acute inflammation is more frequent with MSAF than with clear amniotic fluid, and this relation was observed in particular for WPLI (*p* = 0.008) [10,31].

There were several limitations to our study, including the retrospective nature of the research, the absence of a clinically and histologically negative control group, and the potential effects of therapeutic decisions (e.g., antibiotic therapy) on neonatal outcomes. Additional studies are needed without the bias of wide antibiotics administration. Despite the retrospective nature of the research, we cannot exclude the possibility of selection bias represented by the lack of placental submission in pregnancies in which clinical chorioamnionitis occurred but histological analysis was not performed. Furthermore, there are several common biases that typically accompany a retrospective study, such as the loss of information during the data-collecting process and the inclusion of confounding biases not considered in the analysis. Another limitation is the lack of amniotic fluid analysis to define confirmed TI, as suggested by the NICHD definition. Moreover, since at term intrauterine infection and sterile inflammation can both occur together, amniotic fluid analysis (e.g., white blood cell count, IL-6, glucose concentration) could be used in the future to differentiate these two conditions [32]. In addition, we could not evaluate the independent role of fetal inflammatory involvement in composite adverse neonatal outcomes due to the low incidence of this complication. Another limitation is that we used Redline’s criteria to define placental histology, and these criteria have been recently updated by the Amsterdam Placental Workshop Group Consensus Statement of 2016. This update only recognizes stages 2 and 3 on the maternal side as representing fully developed histologic acute chorioamnionitis, and this has to be specified with or without fetal inflammatory response [33]. Despite using Redline’s criteria in our analysis, maternal stage 1 alone did not contribute to the determining of WPLI in our series; hence, WPLI occurred only with maternal stage 2 and 3 positive placental histology plus fetal inflammatory response.

The strengths of our study are represented by the peculiarity of the investigated topic (triple I in term pregnancy), on which there has been little research published so far, as well as the homogeneity of the neonatal and maternal care management, based on our standardized Maternal–Fetal Unit protocols, which strongly limited the possibility of different treatment and management depending on single-practitioner choices. Other strengths include the high percentage of vaginal deliveries (85%), which allowed us to study the clinical evolution of term labor, as well as the large amount of available histological data.

## 5. Conclusions

The TI criteria seem to be independent and reliable predictors of adverse perinatal complications and of positive histological findings. In the presence of antenatal clinical signs that do not conform with the TI criteria, but which accompany clinical suspicion of IUI, we suggest placental histopathologic examination, since the diagnosis of WPLI can be integrated with clinical presentation due to its independent ability to predict composite neonatal outcomes. Even in cases of TI+, WPLI can improve the diagnostic accuracy of the clinical presentation.

Future research should explore the value of diagnostic–therapeutic protocols inclusive of the TI criteria and fast placental analysis for outcome prediction.

## Figures and Tables

**Figure 1 children-10-01110-f001:**
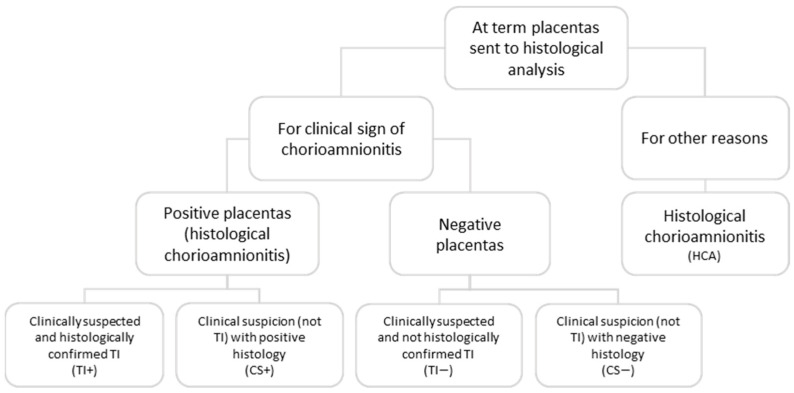
Flow chart describing the study population and the sub-groups based on clinical and histological data.

**Figure 2 children-10-01110-f002:**
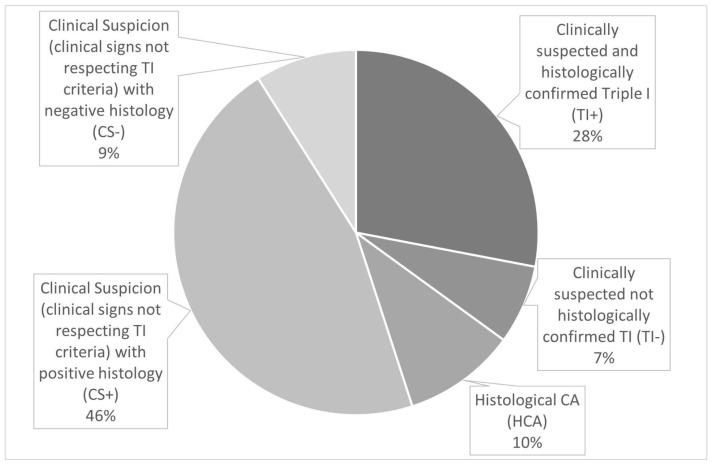
Distribution of the study population based on clinical and histological data.

**Figure 3 children-10-01110-f003:**
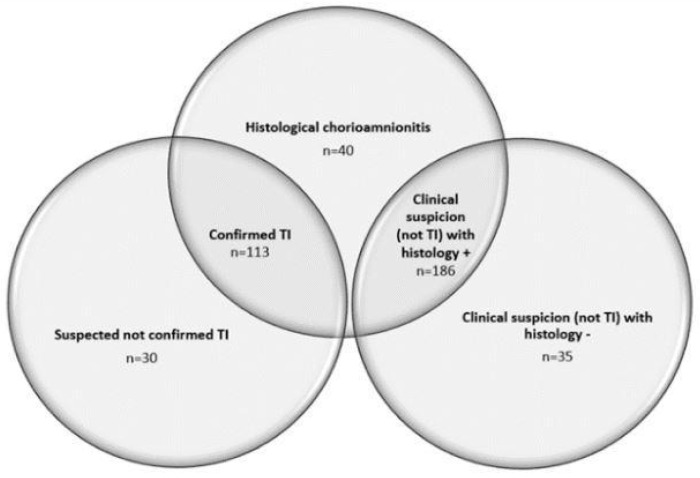
Distribution of diagnostic categories in the study population.

**Table 1 children-10-01110-t001:** Maternal and labor characteristics of the sub-groups. Mean ± SD or % (Nr).

Variable	TI+ (%) m ± s.d. *n* = 113	TI− (%) m ± s.d. *n* = 30	HCA (%) m ± s.d. *n* = 40	CS+ (%) m ± s.d. *n* = 186	CS− (%) m ± s.d. *n* = 35	TOTAL (%) m ± s.d. *n* = 404
Age	32.2 ± 5.4	31.1 ± 6.0	33.7 ± 5.1	33.1 ± 5.2	32.2 ± 5.8	32.7 ± 5.4
Gestational age (weeks + days)	40.1 ± 1	39.2 ± 1	39.5 ± 1	40.0 ± 1	40.1 ± 1	40.0 ± 1
BMI > 35	6.2 (7/113)	6.7 (2/30)	2.5 (1/40)	7.5 (14/186)	14.3 (5/35)	6.4 (26/404)
Nulliparity	75.2 (85/113)	63.3 (19/30)	2.5 (1/40)	54.3 (101/186)	68.5 (24/35)	61.6 (249/404)
PROM	25.6 (29/113)	33.3 (10/30)	35 (14/39)	20.4 (38/182)	17.1 (6/35)	24.0 (97/404)
Positive GBS	31 (35/113)	16.7 (5/30)	26.5 (9/34)	24.7 (45/182)	25.8 (8/31)	26.3 (102/388)
Labor induction	39.8 (45/113)	43.3 (13/30)	12.5 (5/40)	33.3 (62/186)	45.7 (16/35)	34.9 (141/404)
Epidural anesthesia	80.5 (91/113)	76.7 (23/30)	22.5 (9/40)	48.4 (90/186)	51.4 (18/35)	57.2 (231/404)
Use of oxytocin	64.6 (73/113)	56.7 (17/30)	17.5 (7/40)	41.4 (77/186)	40 (14/35)	46.5 (188/404)
Antibiotics during labor	89.4 (101/113)	83.3 (25/30)	35 (14/40)	47.0 (87/185)	48.6 (17/35)	60.5 (244/404)
Meconium-stained amniotic fluid (MSAF)	47.8 (54/113)	40.0 (12/30)	0 (0/40)	58.6 (109/186)	91.4 (32/35)	51.0 (206/404)
Duration of labor (minutes)	448 ± 249	374 ± 231	185 ± 211	329 ± 243	282 ± 222	347 ± 250
Vaginal examinations during labor ≥ 4	93.8 (106/113)	93.0 (27/30)	52.9 (18/40)	78.9 (142/180)	70.0 (21/30)	81.9 (314/383)
Vaginal delivery	65.5 (74/113)	66.7 (20/30)	72.5 (29/40)	68.8 (128/186)	48.6 (17/35)	66.3 (268/404)
Caesarean section	34.5 (39/113)	33.3 (10/30)	27.5 (11/40)	31.2 (58/186)	51.4 (18/35)	33.6 (136/404)

**Table 2 children-10-01110-t002:** Maternal and neonatal adverse outcomes. Mean ± SD % or (Nr).

Variables	TI+ *n* = 113	TI− *n* = 30	HCA *n* = 40	CS+ *n* = 186	CS− *n* = 35	TOTAL *n* = 404
Blood loss > 1000 mL	11.5 (13/113)	13.3 (4/30)	7.5 (3/40)	19.2 (19/186)	5.7 (2/35)	9.4 (38/404)
Postpartum fever	45.1 (51/113)	50 (15/30)	2.5 (1/40)	19.9 (37/186)	17.1 (6/35)	27.2 (110/404)
Postpartum antibiotics	60.2 (68/113)	70 (21/30)	12.5 (5/40)	23.7 (44/186)	25.7 (9/35)	36.4 (147/403)
Endometritis	0.9 (1/113)	0 (0/30)	0 (0/40)	2.7 (5/186)	0 (0/35)	1.5 (6/404)
Maternal hospital re-admission	4.0 (2/49)	0 (0/30)	2.5 (1/40)	3.2 (5/155)	0 (0/35)	2.6 (8/309)
Maternal composite outcome	72.6 (82/113)	76.7 (23/30)	20 (8/40)	36.6 (68/186)	31.4 (11/35)	47.5 (192/404)
Mean birth weight (g)	3415 ± 417	3142 ± 455	2999 ± 494	3254 ± 460	3321 ± 494	3305 ± 468
Apgar score < 7 at 5 min	8.9 (10/113)	6.7 (2/30)	5 (2/40)	11.3 (21/186)	0 (0/35)	8.7 (35/404)
Neonatal CRP > 1 mg/dL	37.5 (33/88)	16.7 (4/24)	10 (1/10)	18.4 (16/87)	100 (15/15)	24.1 (54/224)
Early onset sepsis	3.5 (4/113)	0 (0/21)	0 (0/40)	0 (0/186)	0 (0/2)	1.1 (4/362)
Neonatal antibiotic therapy after discharge	12.4 (14/110)	3.5 (1/29)	0 (0/29)	3.0 (5/168)	0 (0/34)	5.4 (20/370)
NICU admission	2.6 (3/113)	0 (0/30)	2.5 (1/40)	2.7 (5/186)	3.7 (1/27)	2.5 (10/396)
Neonatal composite outcome	73.5 (83/113)	76.7 (23/30)	12.5 (5/40)	28.5 (53/186)	20 (7/35)	42.3 (171/404)

**Table 3 children-10-01110-t003:** Independent predictors of neonatal composite outcome.

Variable	OR	95% CI	*p*-Value
WPLI	2.23	1.41–3.53	0.001
TI+	2.21	1.40–3.47	0.001

**Table 4 children-10-01110-t004:** Diagnostic accuracy of WPLI and TI+ (95% confidence interval).

Variable	Composite Neonatal Outcome	
	Triple I+	WPLI
Sensitivity	56.1% [50.7–61.4]	50.7% [45.3–56.2]
Specificity	84.8% [80.5–88.4]	68.5% [63.2–73.4]
Positive predictive value (PPV)	73.5% [68.4–78.0]	53.4% [47.9–58.8]
Negative predictive value (NPV)	72.1% [67.0–76.7]	66.2% [60.8–71.2]
LR+	3.70 [3.17–4.32]	1.61 [1.33–1.95]
LR−	0.52 [0.44–0.60]	0.72 [0.60–0.87]

## Data Availability

The datasets generated and analyzed for the study are available from the corresponding author upon a reasonable request.

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
