# Peer review of "Intra-Amniotic Inflammation or Infection: Suspected and Confirmed Diagnosis of “Triple I” at Term"

_children, 2023, doi:10.3390/children10071110_

Round 1

Reviewer 1 Report (New Reviewer)

I think this report is meaningful and interesting, because I think the concept of “Triple I” may not yet be widespread and I hear this for the first time by this report. However, the results of this study showed the validity of this new index. So, I have some suggestions as follows.

1: I think the data involved in table 1 and 2 is very large. So, I want you to show important points more clearly, for example by underlines or thick characters.

Antibiotics during labor 89.4, 83.3, Duration of labor 448, 374, Postpartum fever 45.1, 50, Neonatal composite adverse outcome 73.5, 76.7, and so on?

2: I think it may be better to show the results of other VARIABLES (factors in logistic regression analysis) in table 3, though they did not show significant impacts.

Author Response

Dear reviewer, thank you for your suggestions, here the answer to your questions

  1. Thank you for your suggestion, we underlined the main findings in the table 1 and 2.
  2. When we performed the multivariable analysis we considered also these possible counfounding variables: maternal pathology, maternal BMI, parity, induction of labor, oxytocin augmentation of labor, PROM >24 hours, epidural in labor, meconium stained amniotic fluid, but none of them related with neonatal composite outcome in a significative way, than we decided to report only the values of the two independent significant variables.

Reviewer 2 Report (New Reviewer)

1.The inclusion criteria were not comprehensive enough. Infection of amniotic cavity was an important influencing factor of infection-related premature delivery.

2. The“Triple I” proposed by ACOG indicates that tests such as amniotic fluid staining, culture, and so on can also define TI (+) , and the paper confirms that (TI +) or not (TI -) is not accurate based on placental histology.

3.Statistical analysis should be performed on laboratory infection indexes, such as white blood cell count, procalcitonin, and adverse neonatal outcomes, such as necrotizing enterocolitis and intraventricular hemorrhage.

4. The discussion section should be more specific about why TI is a better predictor.

The professional terms need to be clear, and some discussions and language need to be more concise.

Author Response

Dear Reviewer, thank you for your suggestions. Here the answers:

  1. We did not include preterm delivery since the design of the study originated from histologic analysis of at term placentas and our aim was to investigate the clinical variables that could be related or not to histopathological data. We conducted a retrospective cohort study including term pregnancies (≥37 weeks) with clinical suspicion or histological diagnosis of IUI or inflammation. Cases in the study originated from the histological analysis of all the term placentas sent to the Pathology Unit because of clinical suspicion of IUI or for other reasons, as represented in Figure 1 in the paper.
  2. Our data did not include amniotic fluid analysis because we do not perform amniocentisis in case of suspected infection of amniotic cavity at term and only occasionally in preterm. In Italy the most widespread gold standard for the diagnosis of intrauterine infection is placental analysis. Thanks to your suggestion, we added the leak of amniotic fluid analysis as a limit of the study. See also answer 4 to the reviewer 2. Also considering that TI+ is not accurate, this condition is related independently to adverse outcomes and its value need to be evaluated in other prospective studies.

  3. Concerning WBC count, it is already taken into account in the TI definition per se (maternal temperature of 38.0 °C or greater plus any of the following: fetal tachycardia, maternal white blood cell count greater than 15,000 per mm3, definite purulent fluid from the cervical os). Moreover, maternal leukocytosis was one of the possible variables defining the CS + category, so this parameter was already considered even in the case of CS.
    We do not have procalcitonin data, since it is not routinely performed in the case of suspected intraamniotic infection and or inflammation at term of pregnancy. 
    Necrotizing enterocolitis and intraventricular hemorrhage are adverse outcomes typical of preterm newborns, indeed they did not occur in any of our cases. This is the reason why we did not consider those outcomes.

  4. We added a more extensive sentence in the discussion to better explain the concept and our results. Comparing TI+ with CS+, TI+ was more related to WPLI (51.3% vs 34.4%; p-value=0.005) and to neonatal and maternal composite outcome (respectively 45.1% and 25.3%, p-value=0.001; 72.6% and 36.6%; p-value <0.001).

This manuscript is a resubmission of an earlier submission. The following is a list of the peer review reports and author responses from that submission.

Round 1

Reviewer 1 Report

This is a retrospective cohort study including term pregnancies with clinical suspicion and/or histological diagnosis of intra-amniotic infection (IAI) and/or inflammation.

1.      Composite neonatal outcome included Apgar score at 5min <7 and or C-Reactive Protein (CRP) > 1mg/dl and/or positive hemoculture and/or antibiotic therapy at discharge and/or subsequent hospital re-admission. However, only 4 cases were early onset sepsis and none of the newborns were re-hospitalized after discharge. CRP > 1mg/dl is not rare in neonates after birth. No data showed antibiotic therapy at discharge were necessary. The criteria for composite neonatal outcome are not very accurate and the criteria may be not very appropriate.

2.   Sensitivity, specificity, positive predictive value (PPV), negative predictive value (NPV), diagnostic accuraray are not high in Table4. Likelihood ratio (LR) has not been well described in Table4.

3.  Based on the data, it is difficult to make a definite conclusion.

4.  Ethical approval has not been mentioned in the article.

Reviewer 2 Report

The way the investigation has been presented is very complicated and hard to understand for the readers, introducing at the same time many variables which can be very easily mixed-up. Way of presentation should be simplified and presented in less complicated way. It is not clearly stated what were the inclusion and exclusion criteria. The possible bias of the retrospective nature of the study should be discussed and presented as the main limitation.